# Joint contribution of socioeconomic circumstances and ethnic group to variations in preterm birth, neonatal mortality and infant mortality in England and Wales: a population-based retrospective cohort study using routine data from 2006 to 2012

Charles Opondo,[ID] Ron Gray, Jennifer Hollowell, Yangmei Li, Jennifer J Kurinczuk, Maria A Quigley

Policy Research Unit in Maternal Health and Care, National Perinatal Epidemiology Unit, Nuffield Department of Population Health, University of Oxford, Oxford, UK

**Correspondence to**
Dr Charles Opondo;
charles.opondo@npeu.ox.ac.uk

## ABSTRACT

**Objectives** This study aimed to describe the variation in risks of adverse birth outcomes across ethnic groups and socioeconomic circumstances, and to explore the evidence of mediation by socioeconomic circumstances of the effect of ethnicity on birth outcomes.

**Setting** England and Wales.

**Participants** The data came from the 4.6 million singleton live births between 2006 and 2012.

**Exposure** The main exposure was ethnic group. Socioeconomic circumstances, the hypothesised mediator, were measured using the Index of Multiple Deprivation (IMD), an area-level measure of deprivation, based on the mother's place of residence.

**Primary and secondary outcome measures** The primary outcomes were birth outcomes, namely: neonatal death, infant death and preterm birth. We estimated the slope and relative indices of inequality to describe differences in birth outcomes across IMD, and the proportion of the variance in birth outcomes across ethnic groups attributable to IMD. We investigated mediation by IMD on birth outcomes across ethnic groups using structural equation modelling.

**Results** Neonatal mortality, infant mortality and preterm birth risks were 2.1 per 1000, 3.2 per 1000 and 5.6%, respectively. Babies in the most deprived areas had 47%–129% greater risk of adverse birth outcomes than those in the least deprived areas. Minority ethnic babies had 48%–138% greater risk of adverse birth outcomes compared with white British babies. Up to a third of the variance in birth outcomes across ethnic groups was attributable to differences in IMD, and there was strong statistical evidence of an indirect effect through IMD in the effect of ethnicity on birth outcomes.

**Conclusion** There is evidence that socioeconomic circumstances could be contributing to the differences in birth outcomes across ethnic groups.

### Strengths and limitations of this study

► This study was based on a large number of observations, which provided sufficient statistical power to investigate associations across a variety of factors and subgroups.

► The study was population based and sufficiently large to enable individual ethnic groups to be studied.

► We used an area-level measure to quantify the socioeconomic circumstances of individuals, although we also conducted a sensitivity analysis based on an individual-level measure available in a subset of the population.

► Observational data such as these are prone to unmeasured confounding, thus despite the reporting of adjusted estimates, some potential for residual confounding by unmeasured factors remains.

## INTRODUCTION

The rates of adverse birth outcomes, and indeed most other adverse health outcomes, have declined steadily over the last several decades all over the world.[1–3] These trends have been attributed to several factors including improvements in access to healthcare, nutrition, sanitation and living standards, access to birth control, literacy and advances in medical science.[1 4] In England and Wales, the three and a half decades between 1970 and 2006 saw a decline in infant mortality by over 70% from 17.7 to 5.0 deaths per 1000 live births.[5] Trends in preterm birth rates on the other hand have been more variable: although the longer term global trend has been one of a gradual decline, over the

past two decades the rates have levelled off in some parts of Europe[6 7] and even slightly increased in others.[8]

Despite these mostly positive trends, stark disparities in birth outcomes persist across some individual and group characteristics. Previous research has shown that worse socioeconomic circumstances,[9] ethnic minority status,[10] lone parenthood, extremes of maternal age, maternal smoking, obesity and poor access and uptake of health services[11] are all associated with poorer birth outcomes. In particular, the risk of infant mortality is higher in ethnic minority babies,[12 13] as are the risks of neonatal mortality and preterm birth.[10]

At the same time, past and recent patterns of migration in England and Wales and many other parts of Europe have resulted in increasing proportions of births to women from ethnic minority groups.[14 15] For example, in 2010 about 31% of all births in the Eurozone[16] and 25% in the UK[17] were to migrant women, most of whom belonged to minority ethnic groups and had higher fertility rates than the majority white groups.[18] Consequently, there has been an increase in the proportion of the population at risk of adverse birth outcomes, with a disproportionate burden borne by minority ethnic groups. This presents an increasingly significant public health issue which must be addressed[19] if we are to sustain the historical decline in most adverse birth outcomes.

Designing interventions and policies aimed at reducing health disparities requires an understanding of the mechanisms through which heterogeneity of outcomes arises.[20] Considering the role of ethnicity and relative deprivation, previous evidence highlights the role of ethnicity as a predictor of socioeconomic circumstances, with economic and social exclusion cited as examples of possible mechanisms of this association,[21–23] and of socioeconomic circumstances as a predictor of adverse outcomes. This study therefore aimed to describe the variation in risks of adverse birth outcomes across ethnic groups and levels of socioeconomic circumstances, and to explore evidence for the role of socioeconomic circumstances in explaining ethnic variations in birth outcomes. We hypothesised that socioeconomic circumstances were a mediator of the effect of ethnic group on preterm birth and neonatal and infant mortality rates.

## METHODS
### Study design
This was a retrospective cohort study of all singleton live births at gestational age of 22 weeks or more in England and Wales between 2006 and 2012.

### Data sources, acquisition and preparation
Data from the national birth and death registries in England and Wales have been linked since 1993. The Office for National Statistics (ONS) checks these data to minimise inconsistencies. However, these data do not include characteristics such as ethnic group and gestational age. To obtain these characteristics, the registries

data have been further linked to the birth notifications made by midwives and birth attendants using the babies' National Health Service (NHS) numbers. Until 2015, birth notifications were done using a system called the NHS Numbers for Babies (NN4B).[24] The data linkage and its evaluation have been described in detail elsewhere.[24] The ONS provided us with linked data relating to births that occurred between 1 January 2006 and 31 December 2012. Births with implausible values for gestational age, that is, those equal to or greater than 43 weeks, and missing values for birth weight were removed from the linked data set. Additionally, births with birth weight exceeding twice the IQR above or below the median birth weight of the sex-gestation-ethnic group-specific stratum were removed, as they were deemed implausible.[25]

### Outcomes, main explanatory variable and covariates
Outcomes of interest were: neonatal death, defined as a death from any cause occurring within the first 28 days of life; infant death, defined as a death from any cause occurring before the first birthday; and preterm birth, defined as a birth occurring before 37 completed weeks of gestation.[26] Gestational age in these data was intended to be based on last menstrual period, although it is likely that some were actually based on ultrasound estimates.[27]

The main explanatory variable was ethnic group of the baby as recorded in the NN4B notification system, based on the list of ethnic categories used in the 2001 Census in England and Wales. The development of this list has been described elsewhere.[28] Nine categories were derived from those reported, namely White British, Other White, Indian, Pakistani, Bangladeshi, Black Caribbean, Black African, a 'Mixed/Other' group (which included all mixed-ethnicity groups; other Asian groups not discretely classified as Bangladeshi, Indian or Pakistani; other Black groups not discretely categorised as Black African or Black Caribbean; Chinese; and groups recorded as 'other') and a 'Not stated' group.[13] These categories are broadly similar to groupings that have been used in recent studies and reports.[29 30] It is uncertain as to whether the reported ethnic group was actually the baby's and not the mother's or whether it was in fact reported by the mother and not a health professional.[31]

Socioeconomic circumstances were considered a potential mediator of the association between ethnicity and birth outcomes. It was assessed using the 2015 English Index of Multiple Deprivation (IMD)[32] for births in England and the 2014 Welsh IMD[33] for births in Wales. The IMD is a composite index of the relative deprivation of a defined geographical area, usually a homogeneous small area of relatively even size containing approximately 1500 people. The index is computed from a series of indicators and statistics measuring the income, employment, health deprivation and disability, educational skills and training, housing and geographical access to services of the individuals living in an area. For the purpose of this analysis, areas were grouped into deciles of IMD, with the most deprived in the lowest decile and the least deprived

in the highest decile. IMD was available for all births in the linked data set.

An individual-level measure of socioeconomic circumstances, the National Statistics Socioeconomic Classification (NS-SEC), was available for a 10% sample of live births in which information on parental occupation recorded at birth registration was coded by the ONS. The NS-SEC is an occupationally based measure of employment relations and conditions of occupations, conceptually relevant in showing the structure of socioeconomic circumstances and explaining various social phenomena.[34] Individuals are grouped into nine categories, which are ordered starting with employment classes corresponding to better socioeconomic circumstances. To align with the direction of ordering of IMD, NS-SEC was reverse-coded so that the lowest category corresponded to the lowest socioeconomic classification and the highest category corresponded to the highest socioeconomic classification. A household NS-SEC was derived from parental occupation by assigning the higher parental NS-SEC classification to the household.

The linked data set also included covariates such as mother's age, mother's country of birth which was grouped into UK and non-UK for this analysis, birth registration type (married parents, unmarried parents living at the same address, unmarried parents living at different addresses or a sole registrant), year of birth between 2006 and 2012 and sex of child. These variables were included in our analyses as potential confounders of the association between socioeconomic circumstances measured using IMD and birth outcomes. Mother's age and year of birth were modelled as continuous variables and any departures from linear association with the outcomes were explored and fitted.

## Analysis

The slope index of inequality (SII) and the relative index of inequality (RII) were estimated to describe the differences in birth outcomes across IMD.[35 36] The SII is a measure of absolute inequality which describes gradients in outcomes across subgroups with a natural ordering. It is expressed in terms of the difference between predicted outcomes of individuals in the highest versus lowest deprivation decile. The RII is a measure of relative inequality, expressed as the ratio of predicted outcome between the highest versus lowest level of deprivation. SII and RII were estimated using the method suggested in the WHO Handbook on Health Inequality Monitoring,[37] which applied regression analysis—including appropriate modelling of any departures from linearity—to predict risks of birth outcomes in the most and least deprived individuals. CIs for the SII and RII were estimated by bootstrapping. As a sensitivity analysis, we repeated these analyses using household NS-SEC instead of IMD in the 10% sample of live births for which NS-SEC was available.

Two approaches were used to investigate the role of relative deprivation in the association between ethnic group and birth outcomes. First, the amount of variation in birth outcomes across ethnic groups which was explained by socioeconomic circumstances—measured using IMD deciles and modelled by linear regression—was estimated as the relative change in the residual ethnic group-level variance in outcomes as a function of relative deprivation.[38 39] The second approach, which assumed a causal link between ethnic group and IMD, and between IMD and adverse birth outcomes, involved structural equation modelling (SEM) to test an indirect effect through IMD in the effect of ethnicity on birth outcomes. This was based on the method described by Baron and Kenny[40] (figure 1). It involved modelling the unadjusted effect of ethnic group on birth outcomes (path *c*, the *total effect,* not shown in figure 1), and of ethnic group on IMD decile (path *a*), followed by estimation of the effect of ethnic group on birth outcomes (path *c′*) when adjusted for IMD (path *b*). Since the main exposure variable was multicategorical, the *relative* indirect effect of IMD was estimated for each ethnic group, with white British ethnic group as the comparison group,[41] followed by a Sobel test to assess the strength of evidence for an indirect effect.[42] Evidence of an indirect effect in at least one group is required to conclude overall mediation for the effect of the multicategorical exposure.[41]

The relative indirect effect, that is, the proportion of total effect mediated, was calculated as $ab/c$ or equivalently $1-(c'/c)$. A lack of evidence of effect in path *a* or *b* implied the absence of an indirect effect; a lack of evidence of effect in path *c* implied that there was no effect to be mediated. In either case, the relative indirect effect was meaningless, therefore not calculated. Linear, rather than logistic, regression was preferred for modelling all effects despite the birth outcomes being binary variables because explained variance measures in linear models are most intuitive,[43 44] as is the causal decomposition of bivariate associations in linear models, and given that the large number of observations in this linked data set, the effect estimates and SEs from linear regression of binary outcomes are expected to be identical to those from logistic regression.[45] To correct for multiple testing, we used a stringent threshold of $p < 0.001$ when assessing the evidence for direct and indirect effects. Goodness of fit of the SEM was tested using the root mean square error of approximation (RMSEA),[46] the comparative fit index (CFI)[47] and the Tucker-Lewis index (TLI).[48] These indices range in value between 0 and 1. Smaller values of the RMSEA and larger values of both CFI and TLI represent better fit.

All analyses were of complete cases. Data management, manipulation and analyses were performed using Stata V.13.[49]

## Patient and public involvement

A patient and public involvement (PPI) group was formed to advise on the analysis plan, interpretation of results and dissemination plans for this study. We formed the group by contacting organisations who would potentially be interested in this project and asking if they would

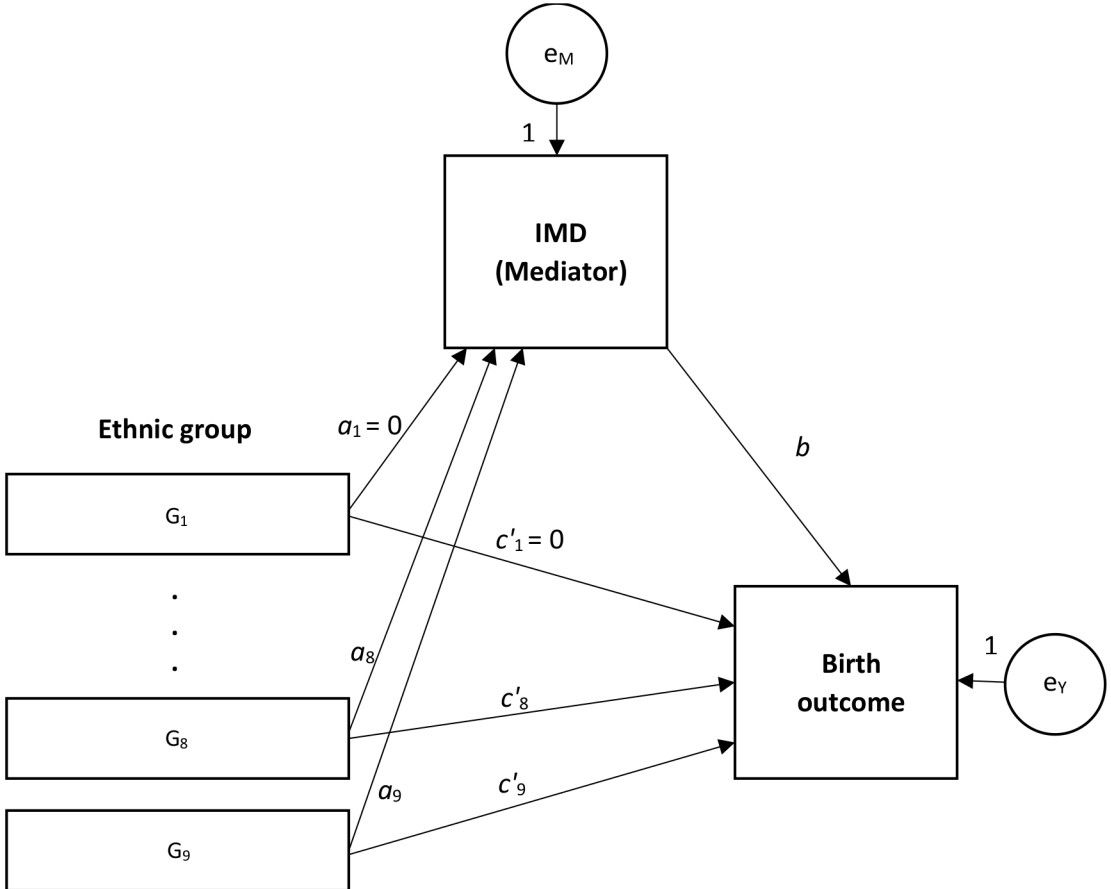

**Figure 1** Mediation model for multicategorical exposure variable. IMD, Index of Multiple Deprivation.

like to join the PPI group. The organisations which sent a representative to our PPI meetings were: Bliss, the Race Equality Foundation, Sands and the Lullaby Trust. In particular, the group provided guidance on ethnic groupings that were used in this study.

## RESULTS

Data on 4 744 666 infants were available in the linked data set. We sequentially excluded 16 695 infants with gestational age of 43 weeks and above, 20 999 infants with missing birth weights and 72 040 infants with implausible birth weights. A total of 4 634 932 infants with complete data in most covariates were included in the main analysis and 463 533 in the sensitivity analysis (online supplementary figure 1). There were 143 infants whose mothers' country of birth could not be coded as UK or non-UK, who were dropped from the adjusted models.

Table 1 is a summary of the characteristics of the infants included in the main analysis. The infants were largely of white British ethnic origin (65%) and mostly born to UK-born mothers (76%). The proportion of births in which the child's ethnicity was not stated was 6.2% overall, although it decreased from 9.9% in 2006 to 3.3% in 2012. There were 2.6% more male than female infants in the data set. The number of births was distributed almost evenly over the 7 years covered by the data set, although there was a gradual year-on-year increase in the number

of births. The mean age of mothers was 29 years. Over half of registrations were by married parents and almost a third were by unmarried cohabiting parents. The rest were either joint registrations by parents living separately or single-parent registrants. There were proportionately fewer infants in the less deprived IMD groups consistent with the distribution of the overall population across the deciles of IMD.

The overall proportion of preterm births was 5.6%. Infant and neonatal deaths were less common, occurring in 3.2 and 2.1 per 1000 live births, respectively. However, more adverse birth outcomes occurred in infants born in deprived areas, overall (table 2) and within ethnic group (figure 2), and a disproportionate number of adverse outcomes occurred among ethnic minority infants (table 2). Additionally, there were disproportionately more ethnic minority infants born in socioeconomically deprived areas. In particular, there were higher proportions of black, Pakistani and Bangladeshi infants in the lower IMD deciles than the overall proportion of infants of these ethnicities, but a lower proportion of the same ethnicities in the higher IMD deciles; the opposite was true for white infants (online supplementary figure 2). The distribution of risks of adverse outcomes across ethnic groups in the 10% sample used for the sensitivity analysis was virtually identical to that in the overall sample (online supplementary table 1).

**Table 1** Characteristics of the 4 634 932 infants born in England and Wales, 2006–2012

| Ethnic group, n (%) | |
|---|---|
| White British | 3 009 231 (64.9) |
| White (other) | 340 526 (7.4) |
| Indian | 132 651 (2.9) |
| Pakistani | 180 269 (3.9) |
| Bangladeshi | 62 948 (1.4) |
| Black Caribbean | 47 505 (1.0) |
| Black African | 154 076 (3.3) |
| Mixed or other ethnic group | 419 970 (9.1) |
| Ethnic group not stated | 287 756 (6.2) |
| Sex of infant, n (%) | |
| Male | 2 377 766 (51.3) |
| Female | 2 257 166 (48.7) |
| Year of birth, n (%) | |
| 2006 | 631 705 (13.6) |
| 2007 | 646 902 (14.0) |
| 2008 | 663 918 (14.3) |
| 2009 | 659 807 (14.2) |
| 2010 | 671 265 (14.5) |
| 2011 | 675 075 (14.6) |
| 2012 | 686 260 (14.8) |
| Mother's age in years, mean (SD) | 29.0 (6.0) |
| Mother's country of birth*, n (%) | |
| UK | 3 507 324 (75.7) |
| Not UK | 1 127 465 (24.3) |
| Birth registration type, n (%) | |
| Registration by married parents | 2 499 063 (53.9) |
| Joint registration by unmarried parents (same address) | 1 398 935 (30.2) |
| Joint registration by unmarried parents (different addresses) | 450 500 (9.7) |
| Sole registrant | 286 434 (6.2) |
| IMD decile, n (%) | |
| 1 (most deprived) | 662 767 (14.3) |
| 2 | 598 259 (12.9) |
| 3 | 541 795 (11.7) |
| 4 | 489 932 (10.6) |
| 5 | 439 270 (9.5) |
| 6 | 422 908 (9.1) |
| 7 | 388 665 (8.4) |
| 8 | 382 585 (8.3) |
| 9 | 366 851 (7.9) |
| 10 (least deprived) | 341 900 (7.4) |
| Preterm birth (<37 weeks' gestation), n (%) | 258 515 (5.6) |
| Neonatal mortality, n (per 1000) | 9638 (2.1) |
| Infant mortality, n (per 1000) | 15 001 (3.2) |

*Excludes 143 infants whose mother's country of birth is unclear or not stated.
IMD, Index of Multiple Deprivation.

**Table 2** Risks of adverse birth outcomes across ethnic groups and socioeconomic circumstances, England and Wales, births 2006–2012

| | Risk of outcome | | |
|---|---|---|---|
| | Neonatal mortality per 1000 | Infant mortality per 1000 | Preterm birth % |
| Ethnic group | | | |
| White British | 1.8 | 2.9 | 5.5 |
| White (other) | 1.6 | 2.5 | 4.6 |
| Indian | 2.4 | 3.6 | 6.0 |
| Pakistani | 4.0 | 6.9 | 6.0 |
| Bangladeshi | 2.7 | 4.4 | 6.3 |
| Black Caribbean | 4.0 | 6.0 | 8.2 |
| Black African | 3.4 | 5.2 | 6.2 |
| Mixed/other | 2.1 | 3.4 | 5.6 |
| Not stated | 2.4 | 3.5 | 5.6 |
| Index of multiple deprivation decile | | | |
| 1 (most deprived) | 2.8 | 4.7 | 6.7 |
| 2 | 2.6 | 4.1 | 6.3 |
| 3 | 2.5 | 3.8 | 6.0 |
| 4 | 2.1 | 3.3 | 5.7 |
| 5 | 1.9 | 2.9 | 5.3 |
| 6 | 1.8 | 2.7 | 5.2 |
| 7 | 1.6 | 2.4 | 5.0 |
| 8 | 1.6 | 2.3 | 4.9 |
| 9 | 1.5 | 2.2 | 4.8 |
| 10 (least deprived) | 1.5 | 2.1 | 4.5 |
| Overall | 2.1 | 3.2 | 55.8 |

Risks are calculated as the number of events in each group divided by the number of infants in the group multiplied by 1000 or 100.

The slope and relative indices of inequality in IMD across birth outcomes are presented in table 3. They show that infants born in the most socioeconomically deprived areas had between 1.47 and 2.29 times the risk of adverse birth outcomes of those in the least deprived areas. The slope and relative indices of adverse birth outcomes across household NS-SEC were similar to those based on IMD: infants born in the most socioeconomically deprived households had between 1.49 and 2.19 times the risk of adverse birth outcomes of those in the least deprived households (online supplementary table 2). Table 3 also shows that a small proportion, between 0.7% and 3.1%, of the overall variance in birth outcomes at ethnic group level was attributable to within-group differences. This implies that most of the variance in birth outcomes was due to between-group differences. Between 25.8% and 35.1% of the variance in outcomes at ethnic group level was explained by IMD in both crude and adjusted regression models.

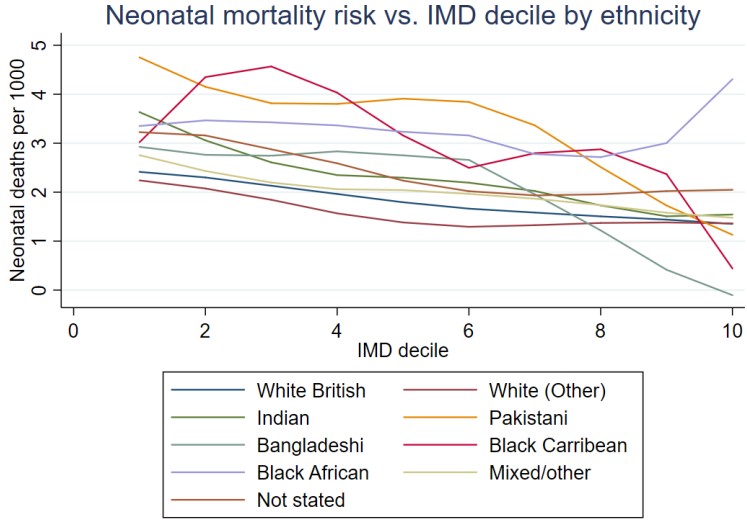

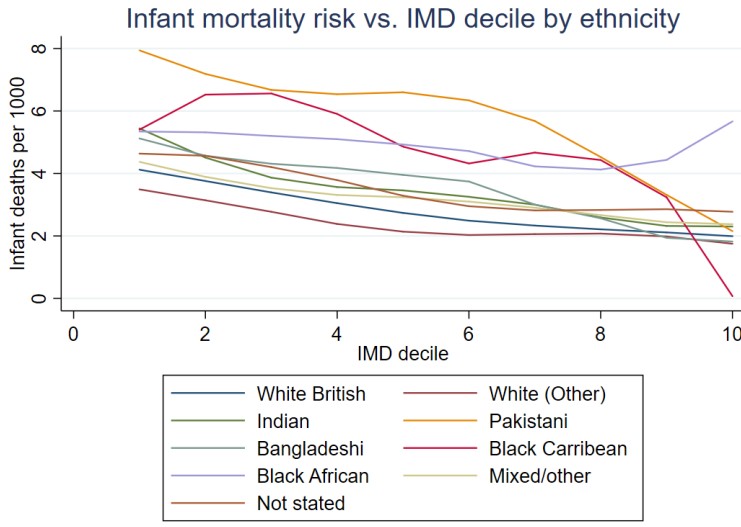

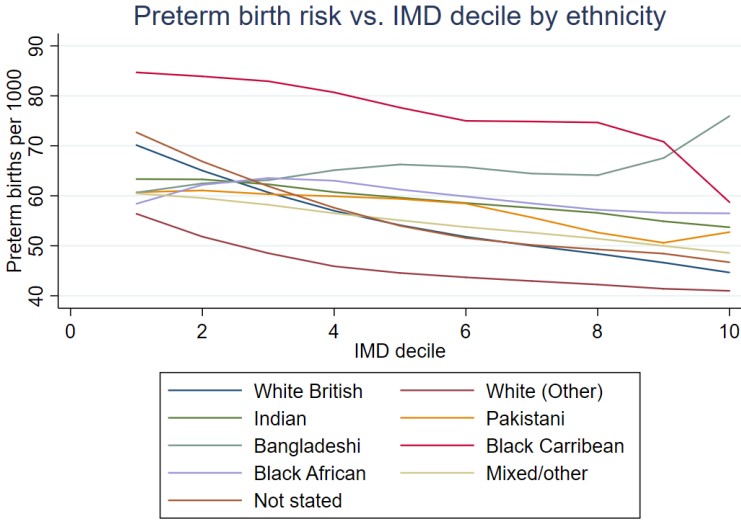

**Figure 2** Distribution of neonatal mortality (top), infant mortality (centre) and preterm births (bottom) across IMD deciles within the ethnic groups, England and Wales, births 2006–2012. IMD, Index of Multiple Deprivation.

**Table 3** Slope and relative indices of inequality (bootstrap 95% CIs), and amount of variation in birth outcomes across ethnic groups explained by IMD, England and Wales, births 2006–2012

|  | Neonatal mortality | Infant mortality | Preterm birth |
|---|---|---|---|
| Index of inequality across IMD |  |  |  |
| Slope index | 1.48 (1.35 to 1.60) | 2.66 (2.50 to 2.82) | 2.15 (2.10 to 2.21) |
| Relative index | 2.05 (1.92 to 2.18) | 2.29 (2.17 to 2.41) | 1.47 (1.46 to 1.49) |
| Proportion of total variance |  |  |  |
| Within ethnic group (crude) | 2.7% | 3.1% | 0.7% |
| Explained by IMD (crude) | 30.1% | 34.0% | 28.2% |
| Explained by IMD (adjusted*) | 25.8% | 27.3% | 35.1% |

*Adjusting for mother's age, mother's country of birth, registration type, year of birth and sex of child.
IMD, Index of Multiple Deprivation.

There was strong evidence of an indirect effect of ethnic group on birth outcomes through socioeconomic circumstances (Sobel test p<0.001). Infants of all ethnic groups except white non-British infants had higher adjusted risks of neonatal and infant mortality than white British infants (online supplementary table 3). The same was true for preterm birth, except there was no evidence of a difference for Pakistani infants, and mixed/other infants had lower adjusted risk of preterm birth than white British infants. The relative indirect effects varied across ethnic groups and across outcomes, from −20.1% for infant mortality in white non-British infants to 78.2% for preterm birth in Black African infants, all relative to white British infants (table 4). Given that the indirect effects were estimated relative to the white British group, negative indirect effect did not imply suppression as it would in absolute mediation models, but a lower indirect effect than the comparison group. The structural equation models were a good fit for the data: in all models the RMSEA was <0.001 and both CFI and TLI were >0.999.

## DISCUSSION

These data from a cohort of babies born in England and Wales between 2006 and 2012 showed poorer birth outcomes among infants of non-white minority ethnic groups. Specifically, ethnic minority infants born over this period had up to twice the risk of adverse birth outcomes of white British infants. These data also showed that among all ethnic groups there was evidence of a gradient in birth outcomes across levels of socioeconomic circumstances, with infants born in the most deprived areas experiencing between 1.47 and 2.29 times the risk of adverse birth outcomes compared with those born in the least deprived areas. Additionally, minority ethnic infants were more likely to be born in deprived areas. Our study aimed to explore the evidence for relative deprivation as the driving mechanism for differences in birth outcomes across ethnic groups. This was based on a causal model in which we hypothesised that relative deprivation was a consequence of minority ethnic status, which was then a cause of poorer birth outcomes.

We found that between a quarter and a third of the total between-ethnic group variance in birth outcomes

**Table 4** Relative indirect effect of ethnic group on birth outcomes, with 95% CIs, England and Wales, births 2006–2012

|  | Neonatal mortality | Infant mortality | Preterm birth |
|---|---|---|---|
| White British | (Comparison group) | (Comparison group) | (Comparison group) |
| White (other) | _ | −20.1% (−35.9 to −4.4) | −8.8% (−9.5 to −8.0) |
| Indian | 14.7% (8.5 to 20.1) | 21.0% (10.4 to 31.6) | 29.7% (19.9 to 39.4) |
| Pakistani | 14.0% (11.3 to 16.7) | 13.9% (12.2 to 15.6) | _ |
| Bangladeshi | _ | 40.0% (24.4 to 55.2) | _ |
| Black Caribbean | 13.4% (10.5 to 16.3) | 16.8% (13.0 to 20.7) | 19.3% (17.0 to 21.5) |
| Black African | 18.6% (14.1 to 23.1) | 22.6% (18.5 to 27.0) | 78.2% (64.2 to 92.2) |
| Mixed or other | _ | _ | _ |
| Not stated | 1.6% (0.8 to 2.3) | 2.5% (0.6 to 4.4) | _ |

Indirect effects are differences relative to the comparison group; they were not calculated in groups and outcomes where there was no evidence of an unadjusted effect of ethnicity on socioeconomic circumstances (path a) or of an adjusted effect of socioeconomic circumstances on birth outcome (path b').

was explained by relative deprivation. This finding, along with the evidence of the association between ethnicity and relative deprivation, and between relative deprivation and adverse birth outcomes, is consistent with a causal link between these factors, although not sufficient proof of causality. We also found strong evidence of an indirect effect through relative deprivation in the association between ethnic group and birth outcomes, with between 13.4% and 78.2% of the risk of adverse birth outcomes in non-white minority ethnic group infants estimated to be due to the higher levels of deprivation in these groups when compared with white British infants. Compared with white non-British infants relative deprivation mediated 8.8%–20.1% of the excess risk of adverse birth outcomes in white British infants. Together, these findings support the hypothesis that relative deprivation is one mediating mechanism through which observed differences in birth outcomes across ethnic groups arise. There may be other mediating mechanisms involving social, demographic or cultural factors, all of which are beyond the scope of this study, and this is further suggested by the finding of incomplete mediation across all groups and outcomes. The public health implication of these findings is that if we were to address the socioeconomic disadvantages borne by ethnic minorities in this population, we could reduce the between-ethnic group disparities in birth outcomes by between a quarter and a third in relative terms.

These conclusions are further supported by previous studies which highlight the associations between ethnicity, relative deprivation and birth outcomes. For example, other studies have shown that birth outcomes and health of infants and children tend to worsen with greater area-level poverty,[50 51] lower household income[52] or in lower social classes,[53 54] all of which are indicators of relative deprivation. Furthermore, previous studies in the UK[23 55] and these data also show a greater propensity for minority ethnic groups to live in socioeconomically deprived areas and to suffer poorer outcomes. However, none of the studies we reviewed investigated the potential mediating effect of socioeconomic circumstances in the association between ethnicity and birth outcomes. Our study expands on previous literature by quantifying the extent to which socioeconomic circumstances contribute to ethnic disparities in birth outcomes.

We hypothesise that the relationship between ethnicity and socioeconomic circumstances, and between socioeconomic circumstances and adverse health outcomes is causal. A number of established findings support this hypothesis of causality and highlight possible mechanisms of effect. For example, language and cultural differences between minority and majority groups can create barriers to accessing or benefiting from information.[56] Minority groups are also more likely to experience various forms of discrimination based on their race, religion or culture, all of which may systematically disadvantage them in accessing services, housing and credit[57]; gaining employment,[58] highlighted by the employment

gap,[59] and rising up the ranks while in employment.[60] In turn, poor socioeconomic circumstances can lead to poor health outcomes through: behavioural mechanisms, for example, tobacco, drug and alcohol use, poorer diet, lower utilisation of health services and contraception[9 61]; reduced access to material resources and safe environments resulting in greater mortality from pollution, poor housing and other health hazards; greater life stress caused by personal circumstances, leading to greater risk of depression, anxiety and poor mental health; with the cumulative effect of all these potential mechanisms over the life course on health and well-being within and across generations.[62]

One limitation of this study is in the use of observational data to explore the causal relationships implied by the mediation model. Observational data are more prone to the occurrence of unmeasured confounding or reverse causality between the main exposure and the outcome, the mediator and the outcome, and between the exposure and the mediator. However, given that the exposure in this study was ethnic group, there were unlikely to be any common causes or reverse causality between it and relative deprivation—the mediator in this analysis—and with birth outcomes. And although there was unlikely to be any reverse causality between relative deprivation and birth outcomes, this association was likely to be confounded by unmeasured factors since both factors may have had external common predictors. In our analysis, we adjusted this pathway for mother's age, mother's country of birth, birth registration type (a proxy measure for parental marital status), year of birth and sex of the child. Although this adjustment did not include factors such as education which may be common predictors of both relative deprivation and birth outcomes,[63] the definition of IMD is broad and encompasses some of these factors, about which we did not have information in this data set—a further limitation of the explanatory power of the study. We also did not have information about important parental factors such as smoking; therefore, there remains some potential for residual confounding. A criticism of the Baron and Kenny approach to the mediation analysis is that it has low statistical power[64]; however, we had the benefit of a large number of observations which greatly improved our ability to detect associations. Another limitation of this study is the use of area-level deprivation to model the socioeconomic circumstances of individuals; this was necessitated by the fact that individual-level indicators of socioeconomic circumstances were not available for the whole sample. This has partly been addressed in the sensitivity analysis in which we have used family NS-SEC derived from parental NS-SEC as an alternative approach to measuring socioeconomic circumstances. Lastly, the ethnic group classifications used in this study may be of limited relevance to other settings.

These findings suggest that further reductions in adverse birth outcomes in England and Wales could be facilitated by interventions that reduce overall social inequality as

this appears to be an important driver of variation in outcomes across ethnic groups. Such interventions must broaden their focus beyond health outcomes alone, and must aim to improve the totality of life circumstances among groups who experience the poorest outcomes.

**Acknowledgements** We thank Bliss, the Race Equality Foundation, Sands and the Lullaby Trust for their helpful involvement in the patient and public involvement (PPI) consultation for this project. We also thank Alison MacFarlane, Hiranthi Jayaweera and Nirupa Dattani for their input during the planning and writing up of this study.

**Contributors** RG, JH, JK and MQ conceived the study and were responsible for data acquisition. CO performed the statistical analysis with support from RG and MQ. CO wrote the first draft, to which RG, JH, YL, JK and MQ contributed and provided feedback during its development. All authors read and approved the final version of the manuscript.

**Funding** This paper reports on an independent study which was funded by the NIHR Policy Research Programme in the Department of Health and Social Care, England (Grant 108/0001).

**Disclaimer** The Department of Health and Social Care was not involved in any aspects of the study, and the views expressed are not necessarily those of the Department.

**Competing interests** None declared.

**Patient consent for publication** Not required.

**Ethics approval** The project was approved by the NRES Committee South Central–Oxford B (Ref 15/SC/0493).

**Provenance and peer review** Not commissioned; externally peer reviewed.

**Data availability statement** The data were provided by the Office for National Statistics (ONS) under a contractual agreement that does not permit the sharing of data. All requests for data access should be made directly to the ONS.

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
