## [Reviewer comments · BMJ Open]

ARTICLE DETAILS

TITLE (PROVISIONAL)	The joint contribution of socioeconomic circumstances and ethnic group to variations in preterm birth, neonatal mortality and infant mortality in England and Wales – a population-based retrospective cohort study using routine data from 2006 to 2012
AUTHORS	Opondo, Charles; Gray, Ron; Hollowell, Jennifer; Li, Yangmei; Kurinczuk, Jenny; Quigley, Maria

VERSION 1 – REVIEW

REVIEWER	Ayesha Sania Columbia University, USA
REVIEW RETURNED	20-Feb-2019

GENERAL COMMENTS	This article contributes to the body of literature describing the difference in risk of adverse birth outcome across ethnic groups and socioeconomic condition in England and Wales. It also evaluates the mediation of effect of ethnicity on birth outcome via socioeconomic condition. The premise of this study is very relevant to the health of infants born to mothers of ethnic minority in England and globally. The study included data from 4.6 million live births, collected from birth and death registries and the NHS. The evaluation of mediation by socio-economic condition is quite valuable and novel. The statistical analysis is well executed, and the paper is easy to follow throughout. I have some minor suggestions for improvement: 1. In abstract, please introduce the concept of IMD in “defined geographic area”. Otherwise it is unclear what is meant by ‘babes born in most deprived areas’.2. Please use a standard for removing implausible birthweight and gestational age combinations instead of using 2X IQR. One widely used standard is published by Alexander et al., based in US population.3. Please mention how missing covariate data was handled in the analyses.4. On table 3, please also include the p values.5. In the discussion section, please include other potential mediators and clarify further why incomplete mediation is likely.
--

REVIEWER	Cindy PADILLA EHESP French School of Public Health, France
REVIEW RETURNED	22-Feb-2019

GENERAL COMMENTS	This manuscript describes the variation in risk of adverse birth outcomes and the most important part is to explore the evidence of mediation by socioeconomic circumstances on the effect of
---

	ethnicity on birth outcomes. While the topic is of interest and timely, authors add to improve some part of the manuscript. Review Comments This manuscript describes the variation in risk of adverse birth outcomes and the most important part is to explore the evidence of mediation by socioeconomic circumstances on the effect of ethnicity on birth outcomes. While the topic is of interest and timely, authors add to improve some part of the manuscript. Suggestions for Authors Introduction: All the ideas are there but this part needs more relevant scientific literature and example according to birth outcomes. The scientific background and rationale is not clear. Page 4 : The paragraph from line 19 to line 31 related to the link between SES and birth outcomes and, ethnic and birth outcomes is very important to understand what your study add in this context. A lot of previous studies made great articles. line 20:Throughout the article, the author uses the term circumstances. I'm more used to seeing the term contextual that circumstances. Line 24 , the author uses independently, I am not agree with that previous articles combine multiples SES factors in their study. So please delete ethnic minority status and independently first. And second, add sentences on previous articles related to ethnic and adverse birth outcomes. Line 30 : You need to add scientific literature to explain and support your sentences, add sentences. Line 33 : This sentence is not clear add a concrete example related to a previous article. Your literature need to be related to birth outcome not health or adult outcomes. Line 47 : the author uses all causes neonatal and infant mortality and in the rest of the article the outcome terms change, be consistent and delete the all causes neonatal Methods : The part related to the setting, locations, dates , data collection is not clear. You need to add more details. Idem for details related to birth outcomes. Page 5 Line 7: We need information and details, which registries ?, from which organization ?, all England ? since when to when ?. Inclusion criteria and exclusion criteria. What information is available in the register ? Please, rewrite the paragraph to be more clearer. Line 21 :Delete Data preparation title and add sentences after "Babies (NN4B)". Line 59 : Delete the sentence at the beginning of the paragraph. Page 6 Line 14 : If I understand this sample of 10% is used for the sensitivity analysis. Please explained. Line 54 : "Between individuals ..." please explained why individuals and not contextual ? Page 7 Line 4: As a sensitivity analysis : please add details on or sensitivity analysis. Results : The author need to rewrite the titles of tables or add sentences to present them. Page 8 Line 15 : You speak about linked dataset . You don't use the same term throughout the article.
--	---

	Line 40: the previous sentence you speak about rates and after risk. Don't present the two sentences in the same paragraph. Results related to table 2: the author speak about risk so we suppose that the risk come from regression but in the table and in the article I saw no p value or other statistical indicators. I suppose that the author describe again and so title need to be clearer and use rates. Line 56 : change your sentence to present the Table 3 according to the regression not only a presentation of the two SES indicators. Table 1 : please present the infant mortality according to your article with per 1000 live births and not %, idem for neonatal mortality. Discussion : your discussion is well done according to your interpretation of your results and hypothesis. But need to be developed according to comparison from similar studies and other relevant evidences. Page 12 Line 8 : Please be careful when you speak about causality. All the criteria are not present to speak about causal relationship Line 31-34 : Delete the sentence related to adults. Lines 25 : Add sentences related to relevant scientific literature according to recent studies in Europe. This paragraph need to be developed.. Line 36 : the sentences need to be developed, previous studies where ? and why ? Page 13 : the last paragraph, the author has to explained the sentence which seems to not be link with the study. Please discuss of your possible external validity (generalisbility) of the study results
--	---

REVIEWER	Jose Guilherme Cecatti University of Campinas Brazil
REVIEW RETURNED	24-Feb-2019

GENERAL COMMENTS	This is a very interesting and elegant manuscript dealing with the role of socioeconomic characteristics and ethnic group for some adverse perinatal outcomes, including neonatal and infant mortality and preterm birth in England and Wales, using population big data and some sophisticated analytical procedures to explore the possible associations between the exposure, mediator and outcomes. I strongly recommend its publication but I would like to see some few points mentioned or discussed in the final version of the manuscript:  1. Is there a concrete reason why only neonatal and infant mortality plus preterm birth were addressed as outcomes? Could you address, with the databases available, other outcomes like neonatal wellbeing (or low Apgar score at 5th min), low birthweight and Small-for-gestational-age rates? This would be really very interesting as well. In additions, did the authors think on the possibility of dealing with a composite variable that would be any of the outcomes used? 2. The ethnic group classification is a matter of concern worldwide, with different countries and cultures using different classifications. This is understandable and an important reflex of this uncertainty is how this information is missing in the majority of studies on
---

	maternal and perinatal health performed by WHO, where real ethnic differences could possibly explain a high proportion of the differences found. Well, while I do understand the use of this standardized way of ethnic classification in UK, it has some limited implications for use outside UK. I mean, the argument is that probably this ethnic classification is not only ethnic but also socioeconomic as well, involving the main mediator focused in this analysis. What are the real ethnic differences between White British and White (others)? The differences between Indian, Pakistanese and Bangladeshi? Where oriental people (Chinese, Japanese, Malasyan, Thai, etc) were classified? I am not proposing to change the classification you used and have available in UK, but I would like to see some of these points discussed better in the discussion session. 3. When presenting the results from Table 1, I think the authors should call attention to the clear trend of declining numbers and proportions of all births from the 1st IMD category (most deprived) to the 10th IMD category (lest deprived) and comment on that.
--	---

REVIEWER	Marie Thoma Department of Family Science, School of Public Health, University of Maryland United States
REVIEW RETURNED	25-Feb-2019

GENERAL COMMENTS	Summary This manuscript examines the association between racial and ethnic groups on infant mortality, neonatal mortality, and preterm birth and how much of the disparities may be explained by socioeconomic status as a mediator. They use linked birth and death registry data from England and Wales between 2006-2012. The authors use a large population-based data source to examine this question. The data was enhanced with indicators of area-level SES and a 10% sample with individual-level SES that was used for sensitivity analyses. The researchers provided a very thorough analysis of this question and used mediation methods, which is a novel approach to examine this question. A strength of this paper is the thorough analysis applied to explore different indices of SES. A limitation is that it is generally already known that some, but not all, of the racial/ethnicity disparity operates through SES. What this study does provide is it quantifies the extent of the disparity across different race/ethnicity groups, which enables a more nuanced discussion of its impact. Introduction p. 4, lines 24-31. While there have been changes to migration patterns and births to ethnic minorities, the birth outcomes have continued to decline in this time. It is unclear what impact this may have had on the racial/ethnic disparities, as migrants tend to be healthier and have better birth outcomes than non-migrants. Unless there has been an increase in disparities, I would rephrase that disparities persist despite improvements in birth outcomes. p.4. This is more of a question than a critique, but would there be changes over time in the extent that SES would mediate the racial/ethnic disparity? If there are changes in populations due to migration, this may also impact time trends in mediation by SES. Methods p. 5. Data sources. More details are needed on the linkages – how they were linked, what % remained unlinked from both the birth and death registries but also the National Health Service data,
---

	which the authors describe contain information on birthweight and ethnic group. What other information was obtained from the linked NHS data – what about gestational age? p. 5, Data sources. Please include the years that were included in the analysis. The abstract says 2006-2012, but this is not in the data source section. p. 5, lines 45-57. The authors describe that these race/ethnicity groupings have been used in other reports and that an advisory group recommended these classifications. Being from the U.S., we think about the classifications differently; however, I trust that this has been explored. The last line about knowing the baby's versus the mother's race – I would think maternal race is actually more important in this case, as it is the mother experiencing the absolute or relative deprivation not the baby. p. 6, lines 1-12. How was IMD applied in your dataset? Was it input by area – what defines the geographic area? The merging of this information may fit better in data source, whereas the description of the measure may fit better in the variable description. Also, was it assumed that IMD in 2014 was the same throughout the study years? p.6, lines 13-15. How was this measure linked to your data source? p.7, lines 19-20. New methods have also been proposed to assess mediation. The discussion should include some of the limitations of the Baron and Kenny approach in line with other newer approaches. p. 7, 44-46. I would not say that the coefficients and standard errors from modeling a binary outcome are identical in linear vs. logistic regression. The former represents the absolute difference, the latter the natural log odds ratio. The properties may be consistent or assumptions satisfied given the large sample size, however. Results Table 4 – how is the negative indirect effect percentage interpreted? Can the relative indirect effect results be interpreted as a measure of public health impact? If we were to address socioeconomic disadvantage in this group, we could reduce the disparity in birth outcomes by X amount? It would be useful to see the magnitude of association for the adjusted total effects, to be able to contextualize the indirect effects findings in Table 4. Discussion You could elaborate a bit more on how your study expands previous literature that addresses the role of SES in racial/ethnic disparities in birth outcomes. You provide mediation analysis, but your study also quantifies the extent of this role for different race/ethnicity groups. In addition, future research could use this approach to examine changes over time, given the changing population structure, in the mediation effect.
--	---

VERSION 1 – AUTHOR RESPONSE

Reviewer (section)	Comment	Response
Editor (general)	Please revise the ‘Strengths and limitations’ section of your manuscript (after the abstract). This section should contain five short bullet points, no longer than one sentence each, that relate specifically to the methods. The results of the study should not be summarised here	We have expanded the ‘Strengths and limitations’ section as suggested.
Reviewer 1 (general)	This article contributes to the body of literature describing the difference in risk of adverse birth outcome across ethnic groups and socioeconomic condition in England and Wales. It also evaluates the mediation of effect of ethnicity on birth outcome via socioeconomic condition. The premise of this study is very relevant to the health of infants born to mothers of ethnic minority in England and globally. The study included data from 4.6 million live births, collected from birth and death registries and the NHS. The evaluation of mediation by socio-economic condition is quite valuable and novel. The statistical analysis is well executed, and the paper is easy to follow throughout.	Thank you for this positive comment. We have responded to your other concerns below.
Reviewer 1 (abstract)	In abstract, please introduce the concept of IMD in “defined geographic area”. Otherwise it is unclear what is meant by ‘babes born in most deprived areas’.	We have clarified that this was an area-based measure of deprivation based on the mother’s place of residence.
Reviewer 1 (methods)	Please use a standard for removing implausible birthweight and gestational age combinations instead of using 2X IQR. One widely used standard is published by Alexander et al., based in US population.	The ‘2 x IQR’ standard has been used to remove implausible birthweights in previously published analyses of this dataset. Although there are other standards that we could have used, to maintain consistency with previous work we would prefer to retain this approach.
Reviewer 1 (methods)	Please mention how missing covariate data was handled in the analyses.	We have specified that this was a complete case analysis in the second-last paragraph of Methods. The linked dataset had 143 observations with a missing mother’s country of birth. They were dropped from the adjusted

Reviewer (section)	Comment	Response
		model. This is indicated in the first paragraph of the results.
Reviewer 1 (results)	On table 3, please also include the p values.	We implemented bootstrapping to generate the confidence intervals in Table 3. This method does not generate p-values, but the confidence intervals provide sufficient information to make statistical inference on the estimates which we have reported. To generate p-values, we would need to implement a computationally intensive permutation test, which would not provide any additional information for inference over that provided by the confidence intervals. From these narrow confidence intervals we can predict that the p-values will be very small.
Reviewer 1 (discussion)	In the discussion section, please include other potential mediators and clarify further why incomplete mediation is likely.	We have now mentioned social, demographic and cultural factors as other potential mediators outside the scope of this study, which may explain the finding of incomplete mediation, in the second paragraph of the discussion.
Reviewer 2 (general)	This manuscript describes the variation in risk of adverse birth outcomes and the most important part is to explore the evidence of mediation by socioeconomic circumstances on the effect of ethnicity on birth outcomes. While the topic is of interest and timely, authors add to improve some part of the manuscript.	Thank you for this positive comment. We have responded to your specific concerns below.
Reviewer 2 (introduction)	All the ideas are there but this part needs more relevant scientific literature and example according to birth outcomes. The scientific background and rationale is not clear.	We have expanded the background and added more literature to the introduction to further clarify the rationale for this study.

Reviewer (section)	Comment	Response
Reviewer 2 (introduction)	The paragraph from line 19 to line 31 related to the link between SES and birth outcomes and, ethnic and birth outcomes is very important to understand what your study add in this context. A lot of previous studies made great articles	We have now explained in the discussion what our study adds to previous ones. Specifically, our study quantifies the extent to which economic circumstances contribute to ethnic disparities in birth outcomes, which previous studies have not attempted.
Reviewer 2 (introduction)	Throughout the article, the author uses the term circumstances. I'm more used to seeing the term contextual that circumstances.	The term 'socioeconomic circumstances' appears to be of standard use in literature on this topic, for example, here , here and here . We would prefer to stick to this terminology since it appears to be widely used and well understood in this context.
Reviewer 2 (introduction)	Line 24 , the author uses independently, I am not agree with that previous articles combine multiples SES factors in their study. So please delete ethnic minority status and independently first. And second, add sentences on previous articles related to ethnic and adverse birth outcomes.	We have removed the word 'independently' from this line. We have also rewritten it to clarify that the associations of the listed factors with poor birth outcomes have not been observed in studies combining multiple SES factors, but separately in separate studies. We have kept 'ethnic minority status' as it is one of the factors we refer to in this line.
Reviewer 2 (introduction)	Line 30 : You need to add scientific literature to explain and support your sentences, add sentences.	We have added a citation in line 30 highlighting the importance of addressing the needs of minority ethnic groups as a public health issue.
Reviewer 2 (introduction)	Line 33 : This sentence is not clear add a concrete example related to a previous article. Your literature need to be related to birth outcome not health or adult outcomes.	We have simplified line 33 to improve its clarity. We have also added a citation to support this statement.
Reviewer 2 (introduction)	Line 47 : the author uses all causes neonatal and infant mortality and in the rest of the article the outcome	We have removed the words 'all cause' to avoid causing confusion.

Reviewer (section)	Comment	Response
	terms change, be consistent and delete the all causes neonatal	
Reviewer 2 (methods)	The part related to the setting, locations, dates , data collection is not clear. You need to add more details. Idem for details related to birth outcomes.	We have re-written the first three paragraphs of 'Methods' to improve the clarity of this information, in response to this and other reviewers' comments.
Reviewer 2 (methods)	Page 5 Line 7: We need information and details, which registries ?, from which organization ?, all England ? since when to when ?. Inclusion criteria and exclusion criteria. What information is available in the register ? Please, rewrite the paragraph to be more clearer.	We have clarified that we refer to the national births and deaths registries in England and Wales. We had edited the first line of the first paragraph of 'Methods' to clarify that we sought all live births at gestational age of 22 weeks or more, in response to the query about the inclusion criteria. 'Data preparation' describes the exclusions. The linkage of registry data have been ongoing since 1993 as indicated in the first line of the second sub-section of 'Methods'.
Reviewer 2 (methods)	Line 21 :Delete Data preparation title and add sentences after "Babies (NN4B)".	We have deleted this title and merged this section into the previous one. We have edited the title of the previous section to include 'preparation'.
Reviewer 2 (methods)	Line 59 : Delete the sentence at the beginning of the paragraph.	We consider that this sentence is helpful to the reader because it states what the potential mediator variable in the analysis is.
Reviewer 2 (methods)	Page 6 Line 14 : If I understand this sample of 10% is used for the sensitivity analysis. Please explained.	Yes – we have now indicated on the last line of the first paragraph of 'Analysis'.
Reviewer 2 (methods)	Line 54 : "Between individuals ..." please explained why individuals and not contextual ?	We have worded this sentence differently to improve clarity: "It is estimated as the difference between predicted outcomes of individuals in..."

Reviewer (section)	Comment	Response
Reviewer 2 (methods)	Page 7 Line 4: As a sensitivity analysis : please add details on or sensitivity analysis.	We have added the details in response to this and the comment about page 6 line 14 above.
Reviewer 2 (results)	The author need to rewrite the titles of tables or add sentences to present them.	We have cited the tables and figures in the text according to the journal requirements.
Reviewer 2 (results)	Page 8 Line 15 : You speak about linked dataset . You don't use the same term throughout the article.	Wherever it applies (e.g. under 'Data sources, acquisition and preparation', 'Outcomes, main explanatory variable and covariates', 'Analysis' and 'Results'), we now refer to the 'linked dataset', except in sections where were are talking about data in a generic sense, such as in the first line of results.
Reviewer 2 (results)	Line 40: the previous sentence you speak about rates and after risk. Don't present the two sentences in the same paragraph	We no longer refer to 'rate' and 'risk' interchangeably in this section.
Reviewer 2 (results)	Results related to table 2: the author speak about risk so we suppose that the risk come from regression but in the table and in the article I saw no p value or other statistical indicators. I suppose that the author describe again and so title need to be clearer and use rates.	Table 2 presents risks as indicated. We have added a footnote to clarify how these risks are obtained. The denominators in the table are not person-times as would be required in the calculation of rates.
Reviewer 2 (results)	Line 56 : change your sentence to present the Table 3 according to the regression not only a presentation of the two SES indicators.	We have edited the sentence to say 'the slope and relative index of inequality in IMD' as stated in Table 3, as suggested.
Reviewer 2 (results)	Table 1 : please present the infant mortality according to your article with per 1000 live births and not %, idem for neonatal mortality.	We have now reported the neonatal and infant mortality risk per 1,000 and preterm birth % all through. We have also edited Tables 2, 3 and the supplementary tables to reflect this change.

Reviewer (section)	Comment	Response
Reviewer 2 (discussion)	Your discussion is well done according to your interpretation of your results and hypothesis. But need to be developed according to comparison from similar studies and other relevant evidences.	Thank you for this positive comment. We have developed this section further according to the specific comments provided by this and other reviewers, as indicated in the responses below.
Reviewer 2 (discussion)	Page 12 Line 8 : Please be careful when you speak about causality. All the criteria are not present to speak about causal relationship	We acknowledge that all criteria for causality are not satisfied: "This finding... is consistent with a causal link between these factors, although not sufficient proof of causality." Nevertheless, we have toned down our assertion and now 'hypothesise' that there is a causal relationship.
Reviewer 2 (discussion)	Line 31-34 : Delete the sentence related to adults.	We have deleted this sentence.
Reviewer 2 (discussion)	Lines 25 : Add sentences related to relevant scientific literature according to recent studies in Europe. This paragraph need to be developed.	We have now added three new citations to recent studies in England, France and The Netherlands, and developed it to clarify the types of outcomes referred to in these studies.
Reviewer 2 (discussion)	Line 36 : the sentences need to be developed, previous studies where ? and why ?	We have clarified that these were previous studies in the UK.
Reviewer 2 (discussion)	Page 13 : the last paragraph, the author has to explained the sentence which seems to not be link with the study.	We have removed the part of this sentence that is not based on the findings of this study.
Reviewer 2 (discussion)	Please discuss of your possible external validity (generalisbility) of the study results	We have added comments to the discussion on generalisability, specifically of the ethnic group categories to other settings.
Reviewer 3 (general)	This is a very interesting and elegant manuscript dealing with the role of socioeconomic characteristics and ethnic group for some adverse perinatal outcomes, including neonatal and infant mortality and preterm birth in England and Wales,	Thank you for this positive comment. We have responded to your other concerns below.

Reviewer (section)	Comment	Response
	using population big data and some sophisticated analytical procedures to explore the possible associations between the exposure, mediator and outcomes. I strongly recommend its publication but I would like to see some few points mentioned or discussed in the final version of the manuscript	
Reviewer 3 (general)	Is there a concrete reason why only neonatal and infant mortality plus preterm birth were addressed as outcomes? Could you address, with the databases available, other outcomes like neonatal wellbeing (or low Apgar score at 5th min), low birthweight and Small-for-gestational-age rates? This would be really very interesting as well. In additions, did the authors think on the possibility of dealing with a composite variable that would be any of the outcomes used?	In this and related papers, we focused on these three well-defined outcomes. We do not have data on neonatal wellbeing or Apgar scores. Analyses of birthweight have been conducted elsewhere. We did not consider a composite outcome variable, as the outcomes reported here are important indicators in their own right, and would have been masked if reported in a composite variable.
Reviewer 3 (general)	The ethnic group classification is a matter of concern worldwide, with different countries and cultures using different classifications. This is understandable and an important reflex of this uncertainty is how this information is missing in the majority of studies on maternal and perinatal health performed by WHO, where real ethnic differences could possibly explain a high proportion of the differences found. Well, while I do understand the use of this standardized way of ethnic classification in UK, it has some limited implications for use outside UK. I mean, the argument is that probably this ethnic classification is not only ethnic but also socioeconomic as well, involving the main mediator focused in this analysis. What are the real ethnic differences between White British and White (others)? The differences between	We have added further explanation in the 'Outcomes, main explanatory variable and covariates' to indicate that other Asian individuals not classified Bangladeshi, Indian or Pakistani (such as Chinese individuals) ended up in the 'mixed/other' group. These ethnic groups are the most common ones in the UK and are also used in UK census data and other analyses involving ethnicity. We have added a comment to the discussion acknowledging the limitation of applying these classifications to other settings with a different ethnic make-up.

Reviewer (section)	Comment	Response
	Indian, Pakistanese and Bangladeshi? Where oriental people (Chinese, Japanese, Malasyan, Thai, etc) were classified? I am not proposing to change the classification you used and have available in UK, but I would like to see some of these points discussed better in the discussion session.	
Reviewer 3 (results)	When presenting the results from Table 1, I think the authors should call attention to the clear trend of declining numbers and proportions of all births from the 1st IMD category (most deprived) to the 10th IMD category (lest deprived) and comment on that.	The pattern of fewer births down the IMD categories (from most deprived to least deprived) reflects the distribution of the population across IMD – more individuals in the child-bearing population fall in the lower categories therefore there would be more births there. We have now added a sentence in the second paragraph of results to indicate this.
Reviewer 4 (general)	This manuscript examines the association between racial and ethnic groups on infant mortality, neonatal mortality, and preterm birth and how much of the disparities may be explained by socioeconomic status as a mediator. They use linked birth and death registry data from England and Wales between 2006-2012. The authors use a large population-based data source to examine this question. The data was enhanced with indicators of area-level SES and a 10% sample with individual-level SES that was used for sensitivity analyses. The researchers provided a very thorough analysis of this question and used mediation methods, which is a novel approach to examine this question. A strength of this paper is the thorough analysis applied to explore different indices of SES. A limitation is that it is generally already known that some, but not all, of the racial/ethnicity disparity operates through SES. What this study does provide is it quantifies the extent of the disparity across	Thank you for this positive comment and helpful suggestions. We have responded to your other concerns below.

Reviewer (section)	Comment	Response
	different race/ethnicity groups, which enables a more nuanced discussion of its impact.	
Reviewer 4 (introduction)	p. 4, lines 24-31. While there have been changes to migration patterns and births to ethnic minorities, the birth outcomes have continued to decline in this time. It is unclear what impact this may have had on the racial/ethnic disparities, as migrants tend to be healthier and have better birth outcomes than non-migrants. Unless there has been an increase in disparities, I would rephrase that disparities persist despite improvements in birth outcomes.	We have now reorganised this paragraph. The first line affirms the persistence of disparities in birth outcomes across groups despite overall improvements.
Reviewer 4 (introduction)	p.4. This is more of a question than a critique, but would there be changes over time in the extent that SES would mediate the racial/ethnic disparity? If there are changes in populations due to migration, this may also impact time trends in mediation by SES.	Yes in theory; if a society evolves over time from one in which an individual's socioeconomic status is strongly determined by their ethnicity to one less so, then the extent to which SES explains differences in outcomes across ethnic groups would decline over this time. However, we don't expect that such momentous changes would occur over a six year period, which is the time covered by these data.
Reviewer 4 (methods)	p. 5. Data sources. More details are needed on the linkages – how they were linked, what % remained unlinked from both the birth and death registries but also the National Health Service data, which the authors describe contain information on birthweight and ethnic group. What other information was obtained from the linked NHS data – what about gestational age?	The data linkage and its evaluation was conducted and described elsewhere. We have indicated this and provided a citation to a detailed description of the linkage, at the end of the first paragraph of 'Data sources, acquisition and preparation'.
Reviewer 4 (methods)	p. 5, Data sources. Please include the years that were included in the analysis. The abstract says 2006-2012,	The first paragraph of 'Methods' under 'Study design' specifies that we sought data from 2006 to 2012. We have also added this to the 'Data

Reviewer (section)	Comment	Response
	but this is not in the data source section.	sources, acquisition and preparation' section.
Reviewer 4 (methods)	p. 5, lines 45-57. The authors describe that these race/ethnicity groupings have been used in other reports and that an advisory group recommended these classifications. Being from the U.S., we think about the classifications differently; however, I trust that this has been explored. The last line about knowing the baby's versus the mother's race – I would think maternal race is actually more important in this case, as it is the mother experiencing the absolute or relative deprivation not the baby.	We have addressed this issue in response to Reviewer 3: these ethnic groups are the most common ones in the UK and are also used in UK census data and other analyses involving ethnicity. Regarding maternal vs. child's ethnicity: in most cases, mother's ethnicity will be the same as that of the baby unless the baby is mixed race and the mother is not. Additionally, because the main measure of deprivation is at area-level and the secondary measure used in the sensitivity analysis is at family-level, our measure of deprivation applies equally to both mother and child.
Reviewer 4 (methods)	p. 6, lines 1-12. How was IMD applied in your dataset? Was it input by area – what defines the geographic area? The merging of this information may fit better in data source, whereas the description of the measure may fit better in the variable description. Also, was it assumed that IMD in 2014 was the same throughout the study years?	We have now indicated that IMD was defined by a geographical area of approximately 1,500 people. We did not define it ourselves, hence the decision to include in the description of variables rather than the data source. IMD in 2014 was generated using the same methodology as in 2010, 2007, 2004 and 2000 (this is explained in the cited report), therefore we expect that it would be unchanged.
Reviewer 4 (methods)	p.6, lines 13-15. How was this measure linked to your data source?	We have now indicated that this measure was derived from parent occupation which was recorded at birth registration.
Reviewer 4 (methods)	p.7, lines 19-20. New methods have also been proposed to assess mediation. The discussion should include some of the limitations of the	We have added some comments on the limitations of the Baron and Kenny approach to the fifth paragraph of the discussion.

Reviewer (section)	Comment	Response
	Baron and Kenny approach in line with other newer approaches.	
Reviewer 4 (methods)	p. 7, 44-46. I would not say that the coefficients and standard errors from modeling a binary outcome are identical in linear vs. logistic regression. The former represents the absolute difference, the latter the natural log odds ratio. The properties may be consistent or assumptions satisfied given the large sample size, however.	We agree. We meant that the effect estimates are expected to be identical (we have checked that they in fact are), not the coefficients , given the reasons you have pointed out. We have clarified this in the text.
Reviewer 4 (results)	Table 4 – how is the negative indirect effect percentage interpreted?	A negative relative indirect effect in this mediation analysis is not interpreted as suppression as would be the case in absolute mediation models, but as a lower indirect effect relative to the comparison group. This is described in the last paragraph of results on page 9 and also highlighted in the footnote of Table 4.
Reviewer 4 (results)	Can the relative indirect effect results be interpreted as a measure of public health impact? If we were to address socioeconomic disadvantage in this group, we could reduce the disparity in birth outcomes by X amount?	Yes – we have added this very helpful interpretation to our discussion of the results.
Reviewer 4 (results)	It would be useful to see the magnitude of association for the adjusted total effects, to be able to contextualize the indirect effects findings in Table 4.	We have now added a supplementary table 3 presenting the unadjusted (total) and adjusted (direct) effects of ethnic group on outcomes, and cited it in the last paragraph of the results.
Reviewer 4 (discussion)	You could elaborate a bit more on how your study expands previous literature that addresses the role of SES in racial/ethnic disparities in birth outcomes. You provide mediation analysis, but your study also quantifies the extent of this role for different race/ethnicity groups. In addition, future research could use this approach to examine changes over	We have added text at the end of the third paragraph of the discussion to state how our study expands previous literature by quantifying the mediating role of SES on ethnic disparities in birth outcomes.

Reviewer (section)	Comment	Response
	time, given the changing population structure, in the mediation effect.	
Formatting amendments	Please provide an 'Article summary' section consisting of the heading: 'Strengths and limitations of this study', and containing up to five (5) bullet points that relate specifically to the study reported. This should be placed after the abstract.	We have expanded the 'Strengths and limitations' section as suggested.

VERSION 2 – REVIEW

REVIEWER	Marie Thoma University of Maryland, School of Public Health United States
REVIEW RETURNED	17-Jun-2019

GENERAL COMMENTS	Summary This manuscript examines the association between racial and ethnic groups on infant mortality, neonatal mortality, and preterm birth and how much of the disparities may be explained by socioeconomic status as a mediator. They use linked birth and death registry data from England and Wales between 2006-2012. The authors submitted revisions that were responsive to my previous comments. I could not access the “response to reviewers,” despite attempts to reach out to the journal. However, it was clear from the manuscript revisions that the main revisions were addressed. I only have a few minor comments that could be further clarified or addressed.  1. The Baron and Kenny approach has its limitations as documented by more recent mediation approaches in causal epidemiology and other fields. Some acknowledgement of this in the limitations of the Discussion is warranted. 2. p.13, lines 3-6. Include citations to studies suggesting these pathways (perhaps some mentioned in lines 31-52 on the same page). In addition, there is little evidence to suggest genetic factors and birth outcomes related to ethnicity per se (or these studies cannot rule out other factors). I suggest removing or providing a solid reference for this. Your discussion on lines 31-52 also implies that social, rather than biologic, factors explain these relationships. 3. p.13, line 60 and p. 14, lines3-8. The sentence about common cause between ethnicity and birth outcomes seems contradictory with the next sentence on unmeasured confounding and external
---

	common predictors. While certain factors would not “cause” ethnicity, ethnicity is only serving as a proxy for a social construct that may represent experiences of deprivation, discrimination, or other factors. Thus, I suggest removing the first sentence regarding common cause and reverse causality and only include the 2nd sentence, which better reflects the mechanism and potential for unmeasured factors to contribute to the analysis. 4. P.8, lines 32-36. What is the rationale for exclusion of births with gestational age of 43 weeks or higher? I understand some very high gestational ages, but 43 weeks is still within reason. In the U.S., reported gestational age of up to 47 weeks is included on birth certificate data given potential imprecision of gestational age measurement. There is generally a 2-week window in which gestational ages may be over or underreported, particularly based on last menstrual period.
--	--

VERSION 2 – AUTHOR RESPONSE

Our responses to the reviewer's comments are as itemised:

1. In response to Reviewer 4 in the previous round of reviews, we had added comments about the limitations of the Baron and Kenny approach in the fifth paragraph of the discussion. As this reviewer could not access the "response to reviewers" document, they may have missed this addition, which was highlighted in the first row in page 8 in the document.

2. We have removed references to genetic factors as recommended. However, we cannot add the citations from further down to this section because those citations refer to associations with social, demographic and cultural factors, not evidence of mediation. In this section, we are merely highlighting that incomplete mediation means there may be other factors beyond the scope of this study that could account for the unexplained mediation.

3. We have edited the sentence about common causes to be clearer that we are referring to common causes between ethnicity and deprivation, and between ethnicity and birth outcomes, as this is an important point to acknowledge as a limitation of mediation models based on observational data.

4. The rationale for excluding births with a gestational age of 43 weeks or higher is that late term pregnancies (41 to 42 weeks) are usually induced before they get to 43 weeks - see <https://www.nice.org.uk/Guidance/CG70>. The reason for induction of these late term pregnancies is that outcomes for the mother and baby are known to get progressively worse the longer a pregnancy goes beyond term. This practice means that gestational ages of 43 weeks would not typically be observed unless (1) the mother refused an induction at this late stage, a very rare occurrence, or (2) the gestational age is incorrect, the more common occurrence in which case makes sense to exclude it as implausible.